# The Effect of Plastic-Related Compounds on Transcriptome-Wide Gene Expression on CYP2C19-Overexpressing HepG2 Cells

**DOI:** 10.3390/molecules28165952

**Published:** 2023-08-08

**Authors:** Matteo Rosellini, Alicia Schulze, Ejlal A. Omer, Nadeen T. Ali, Federico Marini, Jan-Heiner Küpper, Thomas Efferth

**Affiliations:** 1Department of Pharmaceutical Biology, Institute of Pharmaceutical and Biomedical Sciences, Johannes Gutenberg University, Staudinger Weg 5, 55128 Mainz, Germany; mroselli@uni-mainz.de (M.R.); ejlaomer@uni-mainz.de (E.A.O.); neltayeb@unimainz.de (N.T.A.); 2Institute of Medical Biostatistics, Epidemiology and Informatics (IMBEI), University Medical Center of the Johannes, Gutenberg University, 55122 Mainz, Germany; alpoplaw@uni-mainz.de (A.S.); marinif@uni-mainz.de (F.M.); 3Research Center for Immunotherapy (FZI), Langenbeckstraße 1, 55131 Mainz, Germany; 4Institute of Biotechnology, Brandenburg University of Technology Cottbus-Senftenberg, 03046 Senftenberg, Germany; jan-heiner.kuepper@b-tu.de

**Keywords:** cytotoxicity, ecotoxicity, hepatotoxicity, microplastic, RNA sequencing

## Abstract

In recent years, plastic and especially microplastic in the oceans have caused huge problems to marine flora and fauna. Recently, such particles have also been detected in blood, breast milk, and placenta, underlining their ability to enter the human body, presumably via the food chain and other yet-unknown mechanisms. In addition, plastic contains plasticizers, antioxidants, or lubricants, whose impact on human health is also under investigation. At the cellular level, the most important enzymes involved in the metabolism of xenobiotic compounds are the cytochrome P450 monooxygenases (CYPs). Despite their extensive characterization in the maintenance of cellular balance, their interactions with plastic and related products are unexplored. In this study, the possible interactions between several plastic-related compounds and one of the most important cytochromes, CYP2C19, were analyzed. By applying virtual compound screening and molecular docking to more than 1000 commercially available plastic-related compounds, we identified candidates that are likely to interact with this protein. A growth inhibition assay confirmed their cytotoxic activity on a CYP2C19-transfected hepatic cell line. Subsequently, we studied the effect of the selected compounds on the transcriptome-wide gene expression level by conducting RNA sequencing. Three candidate molecules were identified, i.e., 2,2′-methylene bis(6-tert-butyl-4-methylphenol), 1,1-bis(3,5-di-tert-butyl-2-hydroxyphenyl) ethane, and 2,2′-methylene bis(6-cyclohexyl-4-methylphenol)), which bound with a high affinity to CYP2C19 in silico. They exerted a profound cytotoxicity in vitro and interacted with several metabolic pathways, of which the ‘cholesterol biosynthesis process’ was the most affected. In addition, other affected pathways involved mitosis, DNA replication, and inflammation, suggesting an increase in hepatotoxicity. These results indicate that plastic-related compounds could damage the liver by affecting several molecular pathways.

## 1. Introduction

Despite its great importance and utility for our society, plastic poses a huge threat for the oceans [1,2,3,4], marine animals [5,6,7], and the environment [8,9]. Unfortunately, plastic remains in the life cycle for more than 100 years [10]. Through ultraviolet light, seawater, and mechanical action, it degrades to microplastic (smaller than 5 mm) or nanoplastic particles (smaller than 0.1 μm or 1 μm) [11].

Microplastic has recently been detected in human blood [12] and placenta [13], thus confirming its ability to enter our bodies. Indeed, microplastic is not only ingested by fish and other marine animals [14,15,16] but also by humans through food and water [17,18]. Once inside the body, it can cross the gastrointestinal epithelial barrier [19]. Moreover, these particles also enter organisms through respiration and can be detected in the lungs [20]. Once in the blood circulatory system, microplastic can release compounds used as additives for plastic production such as flame retardants, plasticizers, colorants, or antioxidants, which can interact with human cells [21,22].

In the years to come, the amount of plastic is expected to tremendously increase [23]. However, the fate of plastic, microplastic, nanoplastic, and related compounds is still quite elusive once they enter the human body. Some plastic-related chemical pollutants interact with estrogenic receptors [24], while others (i.e., phthalates) appear to be associated with several serious health-related concerns [21]. On the other hand, some chemical pollutants unrelated to plastic (such as aliphatic nitriles or 4-aminobiphenyl) interact with enzymes that are classically associated with drug metabolism [25]. They play an important role in absorption, distribution, metabolization, excretion, and toxicity (ADMETox) [26]. These enzymes are the cytochrome P450 monooxygenases (CYPs). CYPs are hemoproteins responsible for the metabolism of a wide range of substances [27]. Indeed, these proteins play a key role in the biotransformation of both endogenous and exogenous compounds [28]. There are 57 human *CYP* genes, of which only some are involved in the metabolism of drugs and xenobiotics while others are implicated in the metabolism of sterols, vitamins, and other endogenous substrates, or their functions are not yet fully elucidated [29]. These enzymes are expressed at high levels in the liver, specifically in the hepatocytes [30]. Among the various CYPs, CYP2C19 belongs to the most important ones. This enzyme interacts with several molecular partners, including numerous therapeutic drugs such as antiplatelet agents (i.e., clopidogrel), proton pump inhibitors (i.e., omeprazole), or antidepressants (i.e., imipramine, amitriptyline) [31], as well as endogenous compounds such as melatonin and progesterone [32].

Plastic, microplastic, nanoplastic, and their related compounds are increasingly threatening human health [33,34]. However, their possible interactions with CYPs are not yet well understood. In this context, the aim of this study was to investigate several compounds used for plastic production using HepG2 hepatocytes overexpressing CYP2C19. Through in silico screenings, we first identified compounds that could possibly interact with this enzyme from a large library of plastic-related chemicals. In a second step, the most promising candidates were tested with in vitro assays to study their possible toxicity. Finally, using RNA sequencing, we obtained a view of transcriptome-wide gene expression affected by the selected compounds on CYP2C19-overexpressing HepG2 cells. Overall, our comprehensive approach allowed us to determine whether these compounds may have hazardous potential for human health.

## 2. Results

### 2.1. PyRx Screening Analyses

Using the PyRx 0.8 software, we screened more than 1000 compounds downloaded from PubChem associated with plastic. As shown in Figure 1, 46% of the compounds had rather low binding affinities to CYP2C19 ranging between −5.0 and −6.9 kcal/mol. Almost 12% of the selected compounds displayed a high binding affinity to CYP2C19, which were characterized by their binding energy ranging from −8.0 to −12.0 kcal/mol. Based on these parameters, we selected the 70 best compounds with a high affinity to the enzyme for further analyses.

### 2.2. In Silico Binding of Plastic-Related Compounds to CYP2C19

To investigate the binding site of the enzyme more specifically, we performed molecular docking using AutoDock 4.2. We included only compounds (1) with their lowest binding energy (LBE) values smaller than −8.0 kcal/mol, (2) that are commonly used in the plastic industry, and (3) that were commercially available. These criteria skimmed the choice to six compounds (from now on referred to as compounds **1** to **6**, Figure 2). The selected compounds belonged to different classes: compounds **1** and **2** were plasticizers, compound **3** was a UV stabilizer, and compounds **4**–**6** were antioxidants.

The CYP2C19-binding candidates displayed LBE values between −8.14 and −10.38 kcal/mol and predicted inhibition constants (pKi) between 27.73 and 1087 nM. The molecular docking of these compounds was visualized in the LBE conformation. The interactions between the six candidate compounds and the enzyme are shown in Figure 2.

### 2.3. Cytotoxicity

To investigate the effects of the in silico identified compounds on cell viability, we used CYP2C19-overexpressing HepG2 cells (Figure 3). Different concentrations (ranging from 0.003 to 100 μM) were tested. The IC_50_ values of these concentrations are displayed in Figure 3. Compound **1** exhibited an IC_50_ value of 87.99 ± 7.77 μM for the CYP2C19-overexpressing cells. Compound **3** showed a lower IC_50_ value (67.47 ± 2.05 μM). Compounds **4**, **5**, and **6** showed rather comparable IC_50_ values (18.59 ± 0.94 μM, 18.11 ± 0.68 μM, and 20.16 ± 0.06 μM, respectively). Compound **2** did not exhibit any cytotoxic effect.

### 2.4. Transcriptomic Analysis and Deregulated Pathways upon Treatment with Compounds ***4***–***6***

We performed RNA sequencing to analyze the influence of compounds **4**–**6** on the transcriptome-wide gene expression in CYP2C19-overexpressing HepG2 cells.

Differentially expressed genes were determined in the cells treated with compounds **4**–**6** compared to the control cells treated with DMSO. Functional enrichment analysis with clusterProfiler [35] was executed using the gene ontology biological process annotation. Finally, with the GeneTonic software (version 2.4.0) [36], a visual summary of the influenced pathways, from here on referred to as ‘enriched pathways’, was created. These analyses revealed the interactions with the different cellular pathways represented in Figure 4. Interestingly, three pathways (‘cholesterol biosynthetic process’, ‘secondary alcohols biosynthetic process’, and ‘sterol biosynthetic process’) were found to be in common in all the enriched maps.

The pathway enrichment analyses revealed numerous pathways that may help to better explain the detrimental effects of plastic-related compounds in CYP2C19-overexpressing HepG2 cells. This was observed for the pathways related to cell division, e.g., ‘DNA replication’ was suppressed by compounds **4** and **6** and ‘DNA-templated DNA replication’ by compound **4**. Many mitosis-related pathways were also suppressed by compounds **4** and **6** but not by compound **5** (e.g., ‘mitotic sister chromatid segregation’, ‘sister chromatid segregation’, ‘regulation of mitotic sister chromatid separation’, and ‘chromosome organization’). Compound **4** suppressed ‘metaphase/anaphase transition of mitotic cell cycle’, while compound **6** down-regulated ‘nuclear chromosome segregation’, ‘mitotic nuclear division’, and ‘chromosome segregation’. The exposure of cells to compounds **4** and **5** activated cell death mechanisms such as ‘autophagy’ and ‘process utilizing autophagic mechanism’. Interestingly, angiogenesis-related pathways were activated by compound **6**: ‘vasculature development’, ‘blood vessel morphogenesis’, and ‘blood vessel development’. Regarding the metabolic and biosynthetic processes in the cells, ‘response to nutrients levels’ was activated by compound **6**, while many other pathways were downregulated. This was found to be true with all three compounds for ‘cholesterol biosynthetic process’, ‘sterol biosynthetic process’, and ‘secondary alcohol biosynthetic process’. Compounds **4** and **5** suppressed ‘cholesterol metabolic process’ and ‘sterol metabolic process’. Furthermore, compound **5** suppressed the ‘steroid biosynthetic process’, ‘alcohol biosynthetic pathway’, ‘alcohol metabolic process’, and ‘organic hydroxy compound metabolic process’.

We highlighted certain pathways in common among the three compounds selected using signature heat maps generated by the GeneTonic software (version 2.4.0). Figure 5A shows the heat maps of the three compounds vs. DMSO on the most interesting pathway, ‘cholesterol biosynthetic process’. Next, we investigated whether some genes were commonly regulated by the three compounds. Figure 5B highlights these results. Almost all the genes were influenced by the three compounds (twenty-two genes in common), while only one gene was determined to be influenced by compounds **4** and **5**. No genes were in common between compounds **4** and **6** and compounds **5** and **6**. Finally, two genes for compound **4**, three genes for compound **5** and one gene for compound **6** were influenced by only one compound. Then, we created another heat map visualizing the log2-fold change in the genes affected by all three compounds (Figure 5C). Many of the genes in common were highly influenced by all the selected compounds.

In Figure 6A, the heat maps of compounds **4** and **6** on the ‘DNA replication’ pathway are highlighted. The Venn diagram shows seventy-six genes in common between the two selected compounds (Figure 6B). Forty-one genes were exclusively influenced by compound **4**, and nine were uniquely influenced by compound **6**. In the heat map shown in Figure 6C, compound **4** had a higher influence on genes mostly related to the ‘DNA replication’ pathway.

Finally, the two heat maps of compounds **4** and **5** related to the ‘regulation of cytokine production’ are shown in Figure 7A. There were 82 affected genes for compound **4** and 80 for compound **5**. As can be seen in Figure 6B, 54 of these genes were affected by both compounds. These compounds markedly influenced this metabolic pathway.

### 2.5. Predicted Metabolism of Compounds ***4***–***6***

As a final step, we considered whether the metabolization of these compounds by CYP2C19 might influence their binding to this enzyme. As the metabolites of these compounds had not yet been determined, we used the SmartCYP software (version 2.3) [37,38] to theoretically predict the possible metabolites in silico of compounds **4**–**6**. The sites of hydroxylation by CYP2C19 are highlighted in Appendix A. Interestingly, the binding affinities of the predicted metabolites were derived from compounds **4** and **5**, and they were even slightly increased compared to the main compound with only one exception. This can be seen by the comparison of the lowest binding energies (LBE) in Appendix A. For compound **4**, the values ranged from −8.94 kcal/mol (by hydroxylation of C.13) to −10.61 kcal/mol (by hydroxylation of all three selected carbons (C.1, C.13, and C.9)). The LBE value (modifications of C.25) for compound **5** was −9.62 kcal/mol, while the LBE values for the modifications at C.2 and C.10 were −10.96 kcal/mol. For the metabolites of compound **6**, the LBE was −10.48 kcal/mol (hydroxylation of C.11), while −11.00 kcal/mol was the lowest LBE (hydroxylation of C.10 and C.11).

### 2.6. Compounds ***4*** and ***6*** Induce Cell Cycle Arrest in G0/G1 Phase

The results of the flow cytometric cell cycle analysis are shown in Figure 8. The treatment with compounds **4** and **6** revealed an increase in the G0/G1 population at the IC_50_ concentration (74.11 ± 1.98% for compound **4** and 72.30 ± 1.68% for compound **6**, respectively) compared to the untreated cells (66.05 ± 1.66%). In addition, the analysis showed a decrease in the percentage of S-phase cells of the treated compounds (8.05 ± 1.92% for compound **4** and 6.76 ± 3.28 for compound **6**) in contrast to the control (12.83 ± 1.54%). No significant percentage change was detected in the sub-G0 and G2/M phases.

## 3. Discussion

Despite the fact that the release of plastic-related compounds from microplastic has been previously studied [39,40] and the increasing problem of plastic pollution for human health has been recognized [41], their interactions with CYPs have been largely neglected. These enzymes interact with and metabolize a plethora of different substances such as other environmental pollutants (i.e., pesticides or chemicals) [42] and thereby contribute to the maintenance of cellular equilibrium by means of detoxification [26].

Here, we focused on one of the most important CYPs, CYP2C19. As a first step, a library of plastic-related compounds was used. By molecular docking in silico and a growth inhibition assay in vitro, we selected three candidates. We selected compounds **4**–**6** for subsequent analyses, as they bound to this protein with good affinities and were cytotoxic at low IC_50_ values (<21 µM). Considering these results, we studied the possible consequences of their interactions with CYP2C19-overexpressing HepG2 cells at the gene level using RNA sequencing. Previous studies have shown that this approach is feasible, since the gene expression signatures were modulated upon exposure to environmental pollutants [43,44] as well as to hepatotoxic and carcinogenic xenobiotics (e.g., pyrrolizidine alkaloids and polyaromatic hydrocarbons) [42,45]. However, changes in gene expression profiles in CYP-overexpressing cells induced by interactions with plastic-related compounds remain elusive.

Therefore, we investigated transcriptomic alterations in CYP2C19-overexpressing cells treated with the three selected plastic-related compounds. These compounds had some commonly targeted pathways: the ‘cholesterol biosynthetic process’, the ‘secondary alcohol biosynthetic process’ and the ‘sterol biosynthetic process’. However, the pathway found to be the most strongly downregulated for all the three compounds was the ‘cholesterol biosynthesis process’. By analyzing the cholesterol biosynthetic pathway in detail using heat maps of all three compounds and the Venn diagram, it was seen that all three compounds targeted both the same pathway as well as the same genes within it. These included genes that play a critical role in cellular cholesterol homeostasis [46] and whose dysregulation leads to diseases in various organs [47] (such as HMGCR (3-hydroxy-3-methyl-glutaryl-CoA reductase) and DHCR7 (7-dehydrocholesterol reductase)). Other regulated genes, which were also found to show a reduced expression, regulate apolipoproteins, which have been shown to play a key role in cholesterol transport in the blood circulation [48] (APOA1 and APOA4, apolipoprotein A1 and A4).

Microplastics and plastic-related compounds can cross the gastrointestinal epithelial barrier [19] and enter the bloodstream [12]. Subsequently, these particles reach the liver through the portal vein where they interact with CYP2C19-expressing hepatocytes. Plastic-related compounds can accumulate in different organs and compartments, such as liver and adipose tissue [49,50,51]. Therefore, it is plausible that the everyday exposure [52,53] to such compounds and their subsequent accumulation may lead to high local concentrations in the body, especially in the long run, resulting in chronic toxicity [54,55,56].

Cholesterol, which is mainly produced in the liver [57], is a fundamental and ubiquitous molecule in humans. It is involved in numerous biological processes, and its imbalance leads to various diseases and ailments. It is well known that approximately 70% is synthesized in the body (endogenous cholesterol), while the remaining 30% is assimilated through the diet (exogenous cholesterol) [58]. Imbalanced diets or dysregulations in processes affecting the biosynthesis of cholesterol lead to fluctuating cholesterol levels. High cholesterol values in western society are associated with several health problems such as hypercholesterolemia or atherosclerosis [59,60,61]. On the other hand, low concentrations can lead to a different set of issues. Cholesterol is a precursor for the synthesis of substances vital to the body, including steroid hormones, vitamin D, and bile acids [62,63,64]. Due to their lipophilic properties, bile acids facilitate the absorption of lipids and lipid-soluble vitamins. Interestingly, the synthesis of bile acids decreases with age [54,55]. Therefore, the presence of plastic-related compounds could aggravate this lack of bile acid activity in elderly healthy individuals. In fact, these compounds not only hamper the synthesis of endogenous cholesterol but also create imbalances in the absorption of useful substances for the body such as lipids and vitamins. Cholesterol also plays a pivotal role in the integrity of the cell membrane [62,63,65]. A lack of cholesterol production leads to a loss of fluidity and permeability of cell membranes in the presence of plastic-related compounds [66]. This is reflected by a deficit in the regulation of the activity as well as the biophysical properties of numerous ion channels [67]. Moreover, a deficiency of cholesterol in the circulation results in an inadequate distribution of vitamins K and E to vital organs with severe consequences [68].

Furthermore, inflammation-related pathways were activated such as ‘inflammatory response’ for compound **6** or ‘cytokine production’ and ‘regulation of cytokine production’ for compounds **4** and **5**. Inflammatory processes caused by plastics, microplastics [69,70], and their related compounds [71] have been documented. Thus, this response may promote harmful effects on the liver [72] such as the onset of fibrosis and cirrhosis [56].

Our RNA-sequencing analyses additionally showed that compounds **4** and **6** both affected different pathways involved in mitosis (such as ‘mitotic sister chromatid segregation’ or ‘sister chromatid segregation’) and the ‘DNA replication’ pathway. Previously, it has been shown that plastic-related compounds [73,74,75] and environmental pollutants [76,77,78] provoked cytotoxicity and DNA damage, which corroborates our own observations on the cytotoxicity of the compounds. Cell cycle analysis further confirmed this statement. In fact, compounds **4** and **6** accumulated the cells in the G0/G1 phase of the cell cycle. This behavior was also described for other compounds related to plastics (such as bisphenol A) [79]. Other compounds (e.g., gypenoside LI) showed deregulated pathways; similarly, compounds **4** and **6** also arrested the cells in G0/G1 phase [80]. Under healthy physiological conditions, the liver displays efficient regenerative properties [81]. However, plastic-related compounds may reduce this capacity by disrupting mitosis and DNA replication, blocking hepatocytes in G0/G1 phase. Moreover, the suppression of mitosis may cause cell death [82,83]. Regarding compounds **4** and **5**, the activation of pathways involved in autophagy was identified (such as ‘autophagy’ and ‘process utilizing autophagic mechanism’). This is an interesting result as compound **4** has already shown a specific influence on this exact pathway in a previous study [84].

In conclusion, there is no doubt that we are frequently faced with the exposure to plastics, microplastics, and related-compounds [85]. Not only can the main xenobiotics display harmful effects but possibly so can their metabolites [86]. In this context, we determined the common pathways influenced by three selected plastic-related compounds. Interestingly, an augmented toxicity, triggered by simultaneous exposure to several plastic-related compounds, may be due to the discovery that many genes belonging to certain pathways (e.g., the cholesterol pathways) are commonly influenced by the compounds exemplarily selected in this investigation.

## 4. Materials and Methods

### 4.1. Chemicals

Compound 1: dicyclohexyl phthalate (CAS 84-61-7, >99%), compound **2**: diisobutyl phthalate (CAS 84-69-5, >98%), compound **3**: octrizole (CAS 3147-75-9, >98%), compound **4**: 2,2′-methylene bis(6-tert-butyl-4-methylphenol) (CAS 119-47-1, >99%), compound **6**: 2,2′-methylene bis(6-cyclohexyl-4-methylphenol) (CAS 4066-02-8, >97%). Compounds were acquired from TCI Deutschland GmbH (Eschborn, Germany). Compound 5, 1,1-bis(3,5-di-tert-butyl-2-hydroxyphenyl)ethane (CAS 35958-30-6, 96%), was purchased from abcr GmbH (Karlsruhe, Germany).

### 4.2. Cell Lines

CYP2C19-overexpressing HepG2 cells were originated as previously described [87]. Cells were grown in DMEM medium (DMEM, 31966021, Gibco™, Billings, MT, USA) at 37 °C and 5% CO_2_ in a humidified incubator. DMEM media were supplemented with 10% fetal bovine serum (FBS) and with 3 μg/mL blasticidin (ant-bl-05, InvivoGen, San Diego, CA, USA) to preserve the selection of transfected CYP2C19-overexpressing HepG2 cells.

### 4.3. PyRx Screening

More than 1000 compounds associated with plastic production were screened for their interactions with CYP2C19 with the virtual screening tool PyRx (https://pyrx.sourceforge.io) (accessed 16 April 2020). The three-dimensional ligand structures were downloaded from PubChem (NCBI, Bethesda, MD, USA) [88] as standard data files. The crystal structure of CYP2C19 was downloaded from the Protein Data Bank (http://www.rcsb.org/) [89] as a PDB file (PDB code: 4GQS) [90].

### 4.4. Molecular Docking

Molecular docking was run for the compounds with the lowest PyRx binding energies. The binding affinities of the top 70 compounds were read using AutoDock 4.2. The grid box was positioned around the drug binding sites of CYP2C19 with the center of the grid box at x = −99.076, y = −26.072, and z = −63.517 and with the number of grid points (npts) being 100 in x, 104 in y, and 100 in z. Molecular docking was performed with the Lamarckian genetic algorithm with 250 runs and 25 Mio evaluations. The Discovery Studio Visualizer software was used for visualizing protein–ligand interactions. Parts of this analysis were performed using the supercomputer Mogon II and the advisory services offered by Johannes Gutenberg University Mainz (hpc.uni-mainz.de), which is a member of the AHRP (Alliance for High-Performance Computing in Rhineland Palatinate, www.ahrp.info) and the Gauss Alliance e.V. (accessed 25 January 2021).

### 4.5. Resazurin Reduction Assay

Aliquots of 10^4^ CYP2C19-overexpressing HepG2 cells were seeded per well into 96-well plates. Cells were treated with different concentrations of the selected compounds with a range from 0.003 to 100 µM in a total volume of 200 µL for 72 h. Successively, 20 µL/well of resazurin 0.01% *w*/*v* (Sigma Aldrich, Taufkirchen, Germany) was added. The fluorescence intensity was measured with an Infinite M200 Pro-plate reader (Tecan, Crailsheim, Germany). Dose–response curves were engendered for three independent experiments for each compound, and 50% inhibition concentrations (IC_50_) were calculated. The analysis was represented using the Prism 6 GraphPad Software (version 9.5.1) (La Jolla, CA, USA).

### 4.6. RNA Extraction

A total of 24 h before treatment, 5 × 10^5^ CYP2C19-overexpressing HepG2 cells were seeded into 6-well-plates. Cells were treated with the compounds of interest, resulting in a final concentration of IC_50_ calculated beforehand. Control cells were treated with 0.2% DMSO. After 24 h of incubation, the cells were harvested. RNA extraction was performed with an InviTrap^®^ Spin Cell RNA Mini Kit (Invitek Molecular GmbH, Berlin, Germany) according to the manufacturer’s instructions. The cell pellet was lysed with 350 μL of Lysis Solution and treated with β-mercaptoethanol. After DNA removal, 350 μL of 70% ethanol was added, and the sample was applied onto the RNA-RTA spin filter. After several washing steps, RNA was eluted with 60 μL of Elution Buffer R, and the concentration and purity were measured with NanoDrop.

### 4.7. RNA Sequencing

RNA sequencing was performed by StarSEQ GmbH, Mainz, Germany. The quality of the extracted RNA was verified by the company with a 2100 Bioanalyzer system (Agilent Technologies, Santa Clara, CA, USA). After mRNA isolation and library preparation using a NEBNext^©^ Ultra™ II Directional RNA Library Prep Kit (New England Biolabs, Ipswich, MA, USA), RNA sequencing of around 25 Mio PE reads (2 × 12.5 M reads, 2 × 150 nt) was performed with an Illumina NextSeq 2000™ system. Each treatment and control group included two replicates.

### 4.8. Bioinformatics Analysis

The FastQC tool (0.12.0, https://www.bioinformatics.babraham.ac.uk/projects/fastqc/) (accessed 15 February 2023) was used for quality control on the sequencing data. Transcript abundance estimates were computed with Salmon (version 1.5.0) [91] with a transcriptome index generated by GENCODE (version 38) and then summarized to the gene level with the tximeta R package (version 1.16.0) [92]. Exploration, modelling, and interpretation of the expression data followed the protocols defined by Ludt et al. (2022) [93]. Exploratory data analysis was performed with the pcaExplorer package (version 2.24.0) [94]. Differential expression analysis was executed with the DESeq2 package (version 1.38.3) [95], setting the false discovery rate (FDR) cutoff to 0.05. Accurate estimation of the effect sizes (described as log2-fold changes) was performed using the apeglm shrinkage estimator (version 1.20.0) [96]. Further analyses included gene ontology pathway enrichment by topGO (version 2.50.0) [97] using all expressed genes as a background dataset and the ideal package (version 1.22.0) [93] and by clusterProfiler (version 4.6.0) [35] with default settings and the log2-fold change as the input. Then, all pathways with an adjusted *p*-value < 0.05 were chosen [35] for further analyses and processed with the GeneTonic package for visualization and summarization (version 2.2.0) [36]. Gene expression profiles were designed as heatmaps (color-coded standardized Z scores for the expression values after variance-stabilizing transformation) to simplify the comparison across the samples.

### 4.9. Predicted Metabolites

We used SmartCYP [37,38] to predict the possible metabolism sites of the 3 compounds. Then, the ChemDraw software (19.0) was utilized to modify the initial molecules by inserting the hydroxyl group to each of the first 3 predicted ranking sites in different combinations. The binding affinities of the metabolite compounds were calculated using AutoDock 4.2. The grid was placed on the CYP2C19 drug binding sites with the grid center at x = −99.076, y = −26.072, and z = −63.517 and with the number of grid points (npts) being 100 in x, 104 in y, and 100 in z. Molecular docking was performed using the Lamarckian genetic algorithm with 250 runs and 25 million evaluations. The analysis was shown using the Prism 6 GraphPad software (version 9.5.1) (La Jolla, CA, USA).

### 4.10. Cell Cycle Analysis

CYP2C19-overexpressing HepG2 cells were treated with IC_50_ concentrations of compounds **4** and **6** for 24 h. Then, the cells were fixed using 80% cold ethanol and stored overnight at −20 °C. Successively, the cells were washed twice with PBS and resuspended in 1 mL of cold PBS containing 1mg/mL RNaseA (Sigma-Aldrich, Taufkirchen, Germany) and 50 μg/mL PI (Sigma-Aldrich, Taufkirchen, Germany). After 15 min of incubation, the measurements were performed using a BD LSRFortessa™ Cell Analyzer (Becton-Dickinson, Heidelberg, Germany). DMSO-treated cells were used as a negative control.

## 5. Conclusions

We analyzed the interactions of several plastic-related compounds with cytochrome 2C19. Summarizing the results of the three selected compounds (compounds **4**–**6**), a cytotoxic effect could be observed as well as a combined regulation of different gene pathways. Specifically, all three compounds caused significant under-expression in the pathway related to cholesterol biosynthesis. Furthermore, these compounds acted on several pathways related to cell replication (such as mitosis) and inflammation, causing damage to liver cells at different levels. Currently, research regarding the possible toxicity of plastic for humans, specifically concerning the compounds used to produce it, remains very limited. To fully understand their impact on human health, future in vivo research on plastic and related compounds as well as further investigations on related topics are essential. Furthermore, to mitigate the immense problem of plastic pollution, it would be useful not only to conduct scientific research but also to start effectively communicating its findings to the entire population. To this day, plastic pollution is often considered an environmental problem that exclusively concerns the oceans and marine animals; however, day by day, it is increasingly jeopardizing human health as well.

## Figures and Tables

**Figure 1 molecules-28-05952-f001:**
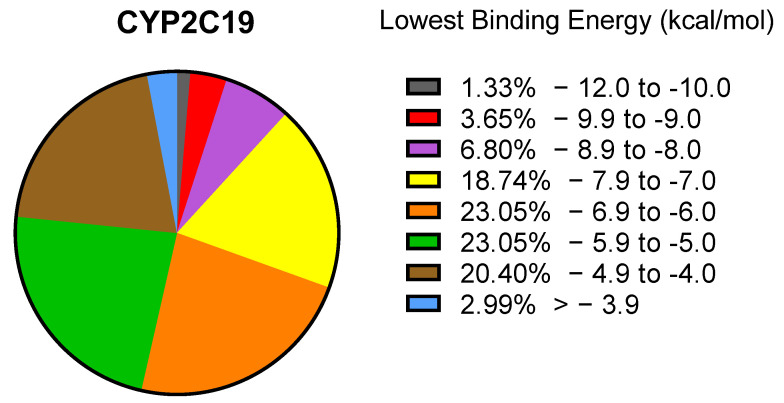
Virtual drug screening using PyRx. The pie chart illustrates the percentage of compounds within a specific range of lowest binding energies to CYP2C19.

**Figure 2 molecules-28-05952-f002:**
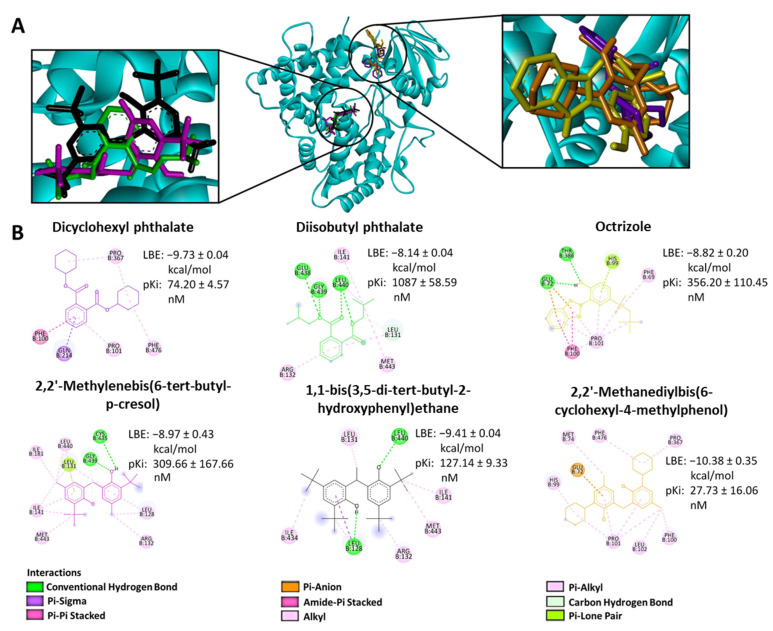
Representation of the binding mode between six selected compounds and CYP2C19. (**A**) Three-dimensional structure of CYP2C19 model (cyan) and the lowest-energy conformation of the selected compounds. (**B**) Two-dimensional representation of the different types of interactions formed between the predicted interactive amino acids of CYP2C19 and the respective selected compounds, as visualized by Discovery Studio Visualizer software (V 21.1.0.20298). The lowest binding energies (LBE) as well as the predicted inhibition constant (pKi) values for each compound with CYP2C19 are shown based on the molecular docking results obtained from AutoDockTools. Chemical structures are shown according to the color code in panel (**A**).

**Figure 3 molecules-28-05952-f003:**
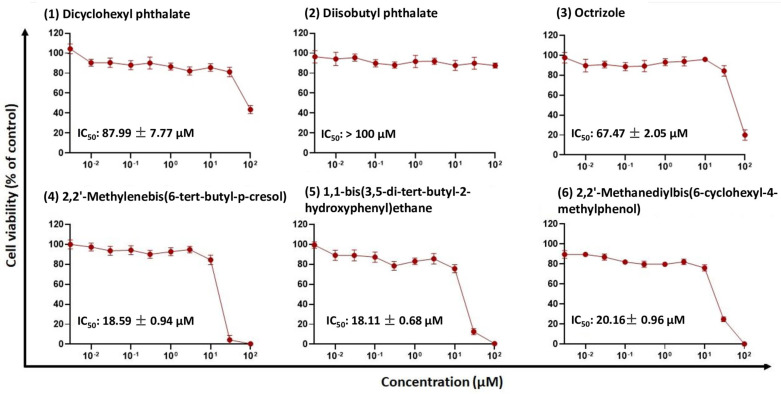
Resazurin assay of cytochrome P450 2C19-overexpressing HepG2 cells treated with different concentrations of the six selected compounds. All experiments were performed in three replicate measurements of mean ± SD.

**Figure 4 molecules-28-05952-f004:**
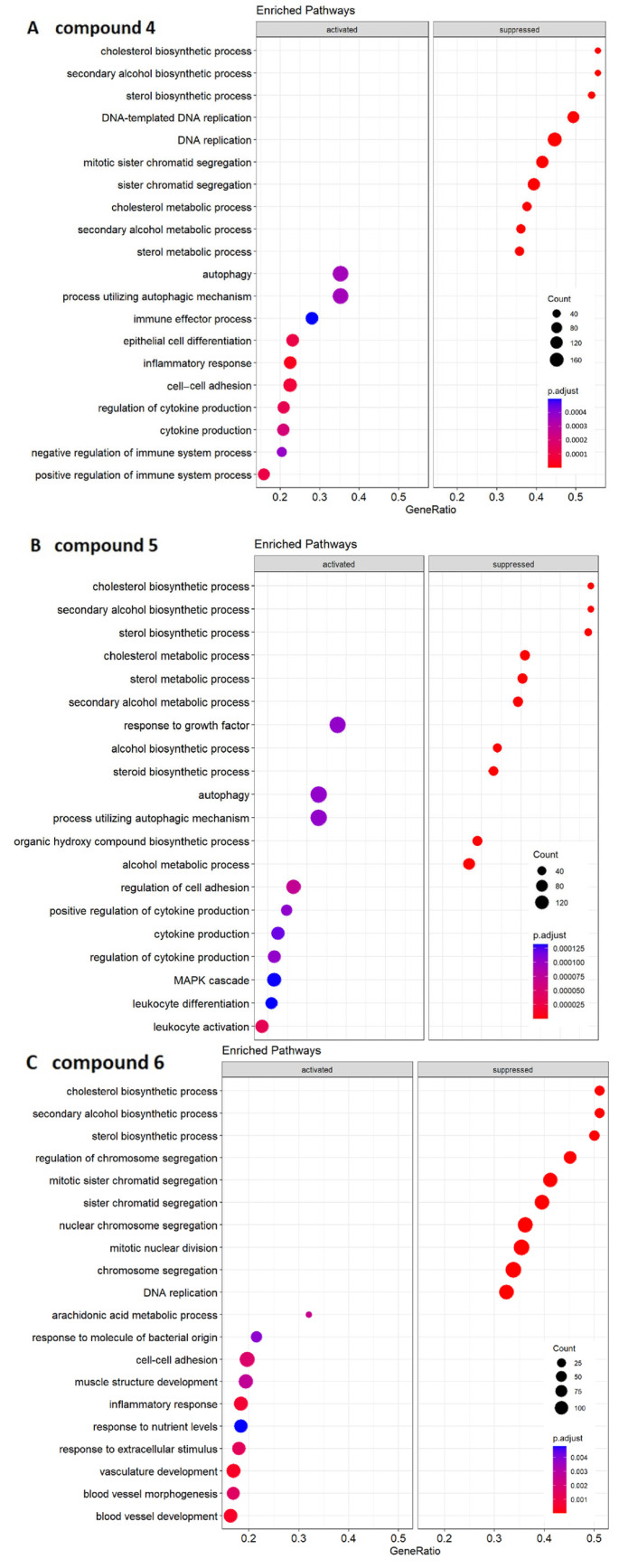
Gene set enrichment analysis showing the top 10 upregulated and the 10 downregulated pathways. (**A**) Enriched pathways referring to compound **4**. (**B**) Enriched pathways referring to compound **5**. (**C**) Enriched pathways referring to compound **6**. Pathways with a positive enrichment score were considered activated and those with a negative score as downregulated. The size of the dots corresponds to the number of genes in the reference gene set. The color of the dots corresponds to the adjusted *p*-value.

**Figure 5 molecules-28-05952-f005:**
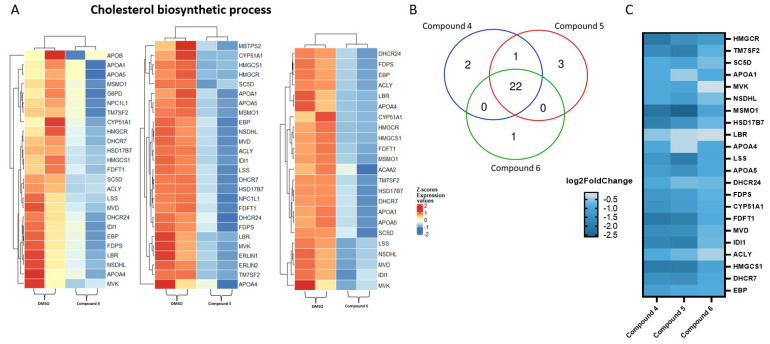
(**A**) Heat maps of differentially expressed genes upon exposure to compounds **4**–**6** related to the ‘cholesterol biosynthetic process’ pathway. The samples selected for comparison always refer to treatment with the compound of interest vs. DMSO. Color-coded standardized z-score expression values after variance-stabilizing transformation was used to simplify the comparison across samples. (**B**) Venn diagram of the genes in common amongst the three selected compounds. (**C**) Heat map showing log2-fold changes in significantly down-regulated genes in common among the three compounds.

**Figure 6 molecules-28-05952-f006:**
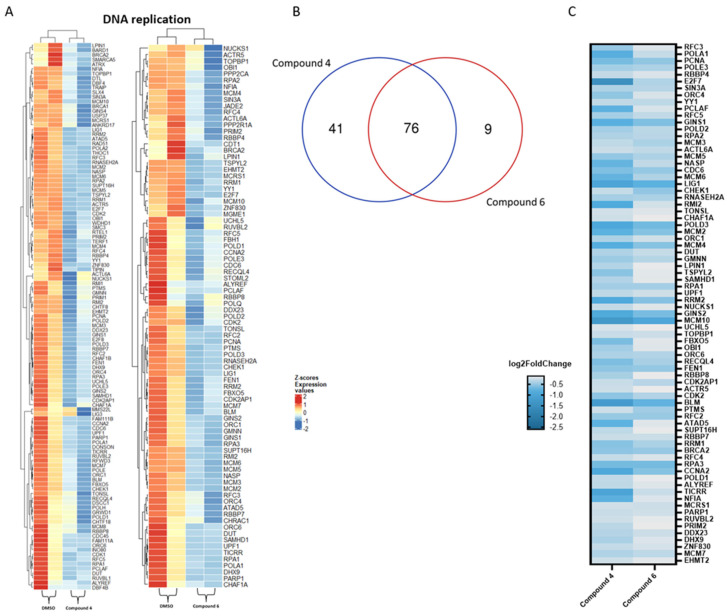
(**A**) Heat maps of differentially expressed genes related to the ‘DNA replication’ pathway upon exposure to compounds **4** and **6**. The samples selected for comparison always refer to the compound of interest vs. DMSO. Color-coded standardized z-scores for the expression values after variance-stabilizing transformation was used to simplify comparison across samples. (**B**) Venn diagram of the genes in common amongst the two selected compounds. (**C**) Heat map showing log2-fold changes in significantly down-regulated genes in common among the two compounds.

**Figure 7 molecules-28-05952-f007:**
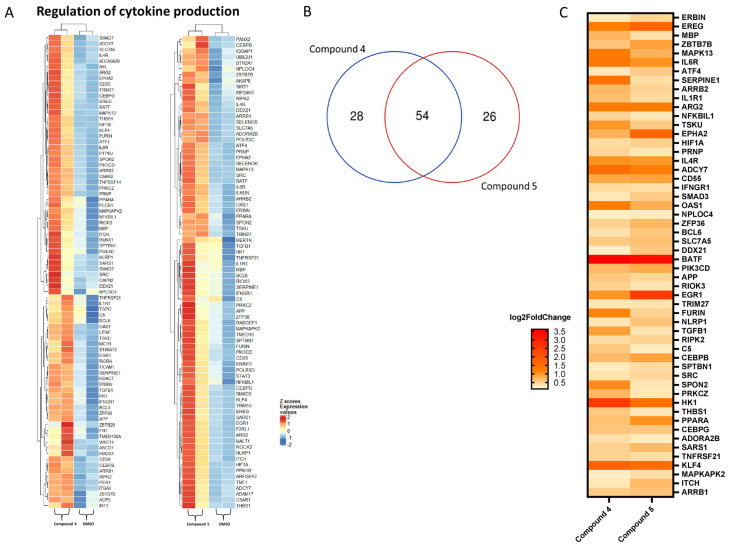
(**A**) Heat maps of differentially expressed genes related to ‘regulation of cytokine production’ pathway upon exposure to compounds **4** and **5**. The samples selected for comparison always refer to the compound of interest vs. DMSO. Color-coded standardized z-scores for the expression values after variance-stabilizing transformation was used to simplify comparison across samples. (**B**) Venn diagram of the genes in common amongst the two selected compounds. (**C**) Heat map showing log2-fold changes in significantly up-regulated genes in common among the two compounds.

**Figure 8 molecules-28-05952-f008:**
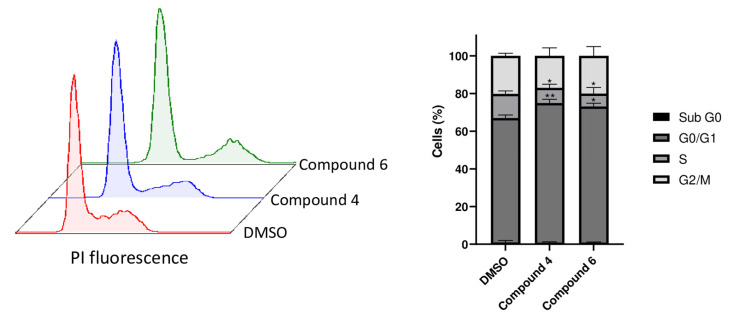
Flow cytometric cell cycle analysis of cytochrome CYP2C19-overexpressing HepG2 cells treated with IC_50_ concentrations of compounds **4** and **6**. The data are represented as mean and SD of three independent experiments. (* *p* < 0.05, ** *p* < 0.01, compared to DMSO-treated cells).

## Data Availability

All sequencing datasets generated for this study have been deposited in the Gene Expression Omnibus (GEO) under accession number GSE237738.

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
