# Peer review of "The Effect of Plastic-Related Compounds on Transcriptome-Wide Gene Expression on CYP2C19-Overexpressing HepG2 Cells"

_molecules, 2023, doi:10.3390/molecules28165952_

Round 1

Reviewer 1 Report

In the submitted manuscript  Rosselinin et al address the issue of the possible impact of compounds derived from microplastics on human health.  Following virtual compound screening and molecular docking of a number of plastic related compounds for CYP2C19 binding the authors identified 6 compounds that predicted to have the lowest energy for binding. Indeed, five of the selected compounds exhibited cytotoxic activity on CYP2C19-overexpressing hepatocytes. Exposure of cells to 3 of the five  microplastic derived compounds led to marked changes  in gene expression as determined by RNA-sequencing. Heat maps of differentially expressed genes led to identification of the pathways altered following exposure to the compounds mainly involved in the cholesterol biosynthetic, DNA replication, and regulation of cytokine production processes.
 The manuscript is a welcomed edition to the increasing interest field of the impact of micro plastics on human health and it is well written. However one issue that is not addressed in the current study  and /or not discussed, is the issue of the concentration of the identified compounds in the human body. In the current study, the cytotoxic activities on cells in culture of the compounds are measured at the low micromolar range. Can these concentrations be attained in the human body and in this case in the liver?  The authors may find some interesting data on the following two publications that are not currently cited in the manuscript :
https://doi.org/10.1016/j.trac.2023.117016 and
https://doi.org/10.1016/j.ebiom.2022.104147

Author Response

Thank you for the revision. In the future, it will be essential to test in-vivo compounds related to plastics as we highlighted in the conclusion. Regarding the concentrations that can be attained, based on both the articles you reported and others in the literature, we have added a new part to the article.

We have also added a new experiment (cell cycle analysis) because we were interested to check if the plastic-related compounds can affect the cell cycle.

Reviewer 2 Report

Article " The effect of plastic-related compounds on transcriptome-wide gene expression on CYP2C19-overexpressing HepG2 cells" (Authors: Matteo Rosellini and at. all). In this study, the possible interactions between several plastic-related compounds and one of the most important cyto-22 chromes, CYP2C19, were analyzed. Plastic, microplastic, nanoplastic, and their related compounds, are increasingly endangering human health. However, their possible interactions with CYPs are not yet well understood. The aim of this study was to investigate several compounds used for plastic production, using HepG2 hepatocytes overexpressing CYP2C19. It is worth noting that the authors correctly placed the emphasis: they study the interaction of monomers with receptors that are used in the production of plastics, and not the plastic materials themselves.

The article is well structured, written in a clear and understandable language, the conclusions are logical, the literature corresponds to the stated topic. The only remark: the text in Figures 2-4 is very small. The request to the authors to try to correct this moment.

I recommend publishing your work.

Author Response

Thank you for the revision. We have modified the figures to make them clearer. For figure 4, we had to change the shape to make it more readable.

We have also added a new experiment (cell cycle analysis) because we were interested to check if the plastic-related compounds can affect the cell cycle.

Reviewer 3 Report

The manuscript "The effect of plastic-related compounds on transcriptome-wide gene expression on CYP2C19-overexpressing HepG2 cells" reports on the study of the possible interactions between several plastic-related compounds and cytochrome CYP2C19 using bioinformatics analysis and cell assays. The authors applied virtual compound screening and molecular docking to more than 1000 commercially available plastic-related compounds and identified candidates which can interact with CYP2C19. The selected six compounds belong to different classes: compounds 1 and 2 are plasticizers, compound 3

is a UV stabilizer, and compounds 4,5, and 6 are antioxidants. Then authors experimentally confirmed their cytotoxic activity on a CYP2C19-overexpressing HepG2 hepatic cell line and studied the effect of three selected compounds on the transcriptome-wide gene expression level by RNA-sequencing. They revealed that three candidate molecules of the antioxidant group, compounds 4, 5, and 6 with high affinity to CYP2C19 in silico, exerted profound cytotoxicity in vitro and interacted with several metabolic pathways of which the ‘cholesterol biosynthesis process’ was most affected (suppressed). Moreover, among other affected pathways were pathways involving DNA replication and mitosis, regulation of cytokine production, and inflammation. Finally, the influence of metabolization of these compounds by CYP2C19 on binding to this enzyme was also considered using the SmartCYP software to theoretically predict the possible metabolites in silico of compounds 4, 5, and 6.

Thus, the presented work's main significance is identifying plastic-related compounds interacted with cytochrome CYP2C19 and affected molecular pathways potentially increasing hepatotoxicity.

This study is well-designed, well-conducted, and well-described. The significance of the studied issues and the study's novelty are substantiated in the Introduction. Methods and Results are described accurately and sufficiently. In general, the presented results support all the conclusions. This manuscript might interest a broad audience of scientists, clinicians, pharmacologists, and ecologists.

I have the following comments on the manuscript.

1. The main remark concerns the validity of the cell model used - CYP2C19-overexpressing HepG2 hepatic cell line. How this in vitro cell model overexpressing CYP2C19 corresponds to in vivo experiments need to be stated.

2. Most of the titles of subsections of the Results section do not reflect the content and the main findings. It is recommended to edit the headings of subsections.

Author Response

Thank you for the revision.

  1. HepG2 cells are cancer cells with mutations that make this cell model only partwise comparable to normal liver tissue. One of the disadvantages of HepG2 cells are that CYP2C19 is not expressed. Therefore, the overexpression of this gene by transfection brings HepG2 cells much closer to the situation in normal liver. Hence the CYP2C19-cells represent an improved cell model which compared better to the in vivo situation (reference 87)
  2. we have modified the results sections to make them clearer.

In addition, we have also added a new experiment (cell cycle analysis) because we were interested to check if the plastic-related compounds can affect the cell cycle.